# Implementing essential diagnostics-learning from essential medicines: A scoping review

**Moriasi Nyanchoka**[1]*, **Mercy Mulaku**[1,2,3], **Bruce Nyagol**[1], **Eddy Johnson Owino**[1], **Simon Kariuki**[1], **Eleanor Ochodo**[1,2]

**1** Centre for Global Health Research, Kenya Medical Research Institute, Nairobi, Kenya, **2** Centre for Evidence-based Health Care, Division of Epidemiology and Biostatistics, Faculty of Medicine and Health Sciences, Stellenbosch University, Stellenbosch, South Africa, **3** Department of Pharmacology, Clinical Pharmacy, and Pharmacy Practice, Faculty of Health Sciences, University of Nairobi, Nairobi, Kenya

* moriasi.nyanchoka@outlook.com

**Data Availability Statement:** All data in this scoping review is included in the tables, figures and appendices.

## Abstract

The World Health Organization (WHO) model list of Essential In vitro Diagnostic (EDL) introduced in 2018 complements the established Essential Medicines List (EML) and improves its impact on advancing universal health coverage and better health outcomes. We conducted a scoping review of the literature on implementing the WHO essential lists in Africa to inform the implementation of the recently introduced EDL. We searched eight electronic databases for studies reporting on implementing the WHO EDL and EML in Africa. Two authors independently conducted study selection and data extraction, with disagreements resolved through discussion. We used the Supporting the Use of Research Evidence (SURE) framework to extract themes and synthesised findings using thematic content analysis. We used the Mixed Method Appraisal Tool (MMAT) version 2018 to assess the quality of included studies. We included 172 studies reporting on EDL and EML after screening 3,813 articles titles and abstracts and 1,545 full-text papers. Most (75%, n = 129) studies were purely quantitative in design, comprising descriptive cross-sectional designs (60%, n = 104), 15% (n = 26) were purely qualitative, and 10% (n = 17) had mixed-methods approaches. There were no qualitative or randomised experimental studies about EDL. The main barrier facing the EML and EDL was poorly equipped health facilities—including unavailability or stock-outs of essential in vitro diagnostics and medicines. Financial and non-financial incentives to health facilities and workers were key enablers in implementing the EML; however, their impact differed from one context to another. Only fifty-six (33%) of the included studies were of high quality. Poorly equipped and stocked health facilities remain an implementation barrier to essential diagnostics and medicines. Health system interventions such as financial and non-financial incentives to improve their availability can be applied in different contexts. More implementation study designs, such as experimental and qualitative studies, are required to evaluate the effectiveness of essential lists.

**Funding:** This study was funded by UK National Institute for Health Research (NIHR) Global HPSR developmental award (130222 to EO). EO, MM, BN, EJO's time was funded under the UK MRC African Research Leaders award (MR/T008768/1). This award is jointly funded by the UK Medical Research Council (MRC) and the UK Foreign, Commonwealth & Development Office (FCDO) under the MRC/FCDO Concordat agreement and is also part of the EDCTP2 program supported by the European Union. The funders had no role in study design, data collection and analysis, decision to publish, or preparation of the manuscript.

**Competing interests:** The authors have declared that no competing interests exist.

## Introduction

Access to diagnostic tests is key to achieving Sustainable Development Goals (SDG) 3.8. and Universal Health Coverage (UHC) [1, 2]. Insufficient access to essential in-vitro diagnostics is a major global health challenge, and nearly half (47%) of the global population have little to no access to diagnostics [3]. The scale and scope of this challenge contribute to delays in diagnosis and initiation of appropriate treatment compromising health outcomes, especially in Africa [3–13].

The World Health Organization (WHO) published the first model list of Essential In Vitro Diagnostics (EDL) in 2018 [14] to guide the selection and prioritisation of essential diagnostics according to national needs. It complements the WHO Essential Medicines List (EML) and links medicines with diagnostic tests to advance the UHC [11, 15]. It paves the way toward improved healthcare delivery and ultimately better patient outcomes by promoting greater equitable access to quality and affordable diagnostics at all levels of the healthcare delivery system [16]. Countries need to adopt and develop national lists that suit their national or regional needs, disease burdens, and health system capacities to ensure their impact on healthcare practice and patient outcomes [2, 5, 17]. To date, the WHO has published three EDL model lists [14, 18, 19]. The first WHO EDL contained 113 tests and was updated by WHO in 2019 to include nine additional tests for non-infectious diseases. The 2020 list had more other tests, including pandemics such as Covid-19.

The WHO published the first EML about 45 years ago, in 1977 This established initiative has been updated biannually since 1977, with the latest 22$^{nd}$ version updated in September 2021 [20]. Though the adaption of WHO EML to National Essential Medicines Lists (NEMLs) has been broad in Africa, numerous challenges continue to blunt its impact, including persistent inadequate and inequitable access to medicines [21–24]. Lessons learned in implementing the established WHO EML may shed light on implementation considerations of the WHO EDL and guide the development of practice tools to support the broader adoption of the WHO EDL in Africa.

The objective of this scoping review was to map evidence on the implementation of the WHO's essential lists in African countries to guide the effective implementation of the new WHO EDL.

## Methods

A protocol of our review can be found in the Open Science Framework [25] with deviations from the protocol listed in the Appendix (S1 Appendix).

We conducted this review according to the Joanna Briggs Institute guidelines for scoping reviews [26] and adhered to the Preferred Reporting Items for Systematic Reviews and Meta-Analyses extension for Scoping Reviews (PRISMA-ScR) [27] checklist recommended for scoping reviews (S1 Checklist).

### Eligibility criteria

**Types of studies.** We included EDL studies published in English after the launch of the first EDL in 2018. However, given the vast number of studies for the 40-year EML initiative, we included EML studies published in 2010 and after to get a recent representative sample that would inform the implementation of the EDL. If data saturation were not achieved in this sample, we would look to studies published before 2010. We included primary experimental and observational studies and primary qualitative studies. We excluded study protocols, literature reviews, systematic reviews, scoping reviews, book chapters, personal opinions papers, editorials, and conference abstracts with insufficient information. We, however, excluded all

conference abstracts and editorials on EML due to the vast number of full-text studies and the high likelihood of data saturation.

We selected eligible studies guided by the Population-Concept-Context (PCC) framework designed by the Joanna Briggs Institute [28], commonly used to focus research questions for scoping reviews as detailed below:

**Population.** We included articles reporting on the provision of essential medicines and diagnostic tests (as defined by authors or by the WHO Criteria) [19, 20] to human populations. We did not limit our review to any disease condition.

**Concept.** We included articles that discussed the implementation of the WHO essential lists, defined in our review as the adoption and adaptation of WHO essential lists by individual WHO member states to address national priority healthcare needs and gaps in the health systems. We also included articles that evaluated interventions used to enhance or enable the implementation or uptake of the essential lists.

**Context/settings.** We included all studies conducted in all health care settings or levels in Africa. Due to the high likelihood of data saturation, we restricted EML studies to a convenient sample of those conducted in single countries. Such studies were likely to give rich data about implementation considerations in one setting or context.

## Outcomes

Our outcomes of interest were:

- Types of study designs about the implementation of the EDL and EML.

- Themes about barriers and enablers of EDL and EML.

## Information sources and search

A systematic literature search was conducted up to May 2021 without date restrictions. We searched several electronic databases: *Ovid MEDLINE, Embase, CINAHL, Web of Science, African Index Medicus, Cochrane Central Register of Controlled Trials, SCOPUS, and Health system evidence for eligible studies*. An example of the search strategies *MEDLINE* can be found in the S2 Appendix. The literature search was complemented by scanning the reference lists of included studies. The references were exported to an EndNote database following the literature search, and the duplicates were removed.

## Study selection

We uploaded references compiled using *Endnote* to Covidence [29], a web-based systematic review software platform. We first screened titles and abstracts for potentially eligible articles and then screened full texts of the articles where available. Independent reviewers (MN, BN, EJO, MM) screened all titles, abstracts, and full-text articles in duplicate for eligibility. We resolved disagreements through consensus in consultation with a senior reviewer (EO). Articles that met the inclusion criteria following a full-text review were selected for data extraction.

## Data items and data charting process

We conducted data extraction using the google form platform developed a *priori*. We piloted it with 10% of the included studies by two reviewers to ensure the accuracy of the form and consistency of the extracted content. After completing the pilot tests, the research team held a meeting for feedback and discussion on discrepancies. Following the consensus, we updated the form before extracting data from the included studies.

We used the Supporting the Use of Research Evidence (SURE) framework suitable for qualitative data extraction (S2 Appendix) [30]. The SURE framework provided a comprehensive list of possible factors (barriers and enablers) that we used to systematically describe the implementation of adopted WHO essential lists. These factors are categorised into five groups: recipients of care, providers of care, other stakeholders, health systems, and social and political constraints (S2 Appendix) [30]. MN, MM, BN, and EJO independently extracted relevant phrases that corresponded to the SURE framework in duplicate. Bibliographic information was also extracted, including first author, year of publication, study title, study type or design, and country of study.

We further analysed the extracted phases, categorised them as barriers and enablers, and coded each phrase with one or more relevant SURE framework codes. The 34 applied codes were grouped into themes and analysed using Microsoft Excel version 16.66. We also explored the frequency of the applied SURE framework codes and used them to calculate the proportion of total themes per code. We resolved disagreements in data extraction through consensus in consultation with a senior reviewer (EO).

## Quality assessment

According to the JBI manual for scoping reviews, quality assessment is not mandatory for scoping reviews but can be applied depending on the nature of the review [26]. In our context, knowing the quality of existing study designs was essential to our appraisal of the available evidence about implementing the EDL and EML. We assessed the methodological quality of all included studies using the Mixed Method Appraisal Tool (MMAT 2018) [31]. The tool is grouped into five categories of study designs: qualitative designs, quantitative randomized controlled trials, quantitative non-randomized, quantitative descriptive, and mixed methods. The appraisal led to an overall methodological quality rating varying from unclassified, 0% (no quality), 20% (low quality), 60% (moderate quality), 80 (considerable quality), and 100% (high quality) for each study. Not all eligible studies provided sufficient information to appraise quality using the MMAT. A study was categorised as unclassified if it was a report or study that did not provide adequate information for MMAT appraisal. We incorporated information for quality assessment into the data extraction form and piloted it in 10% of the included studies. Quality ratings were not used to include or exclude studies but to describe the overall quality of the available evidence about the implementation of the essential lists.

## Synthesis of results

Results were coded and synthesised using thematic content analysis [32]. One reviewer (MN) performed thematic analysis and was verified by a senior reviewer (EO). We used established themes based on the SURE framework [30] to summarise the results descriptively and graphically.

## Results

### Search results

Our search yielded 3813 records, nine of which were duplicates Fig 1. Of the remaining 3804 citations screened, 2259 did not meet the inclusion criteria. A further 1373 citations were excluded at full text review, as they did not meet the inclusion criteria for full text review based on year of publication (n = 523), not focussing on implementation of WHO essential lists (n = 324), ineligible article type (n = 265), ineligible context (n = 102), multi-country studies about EML (n = 67), no full text availability (n = 52), duplicates (n = 21), non-English publication language (n = 16), animal studies (n = 3).

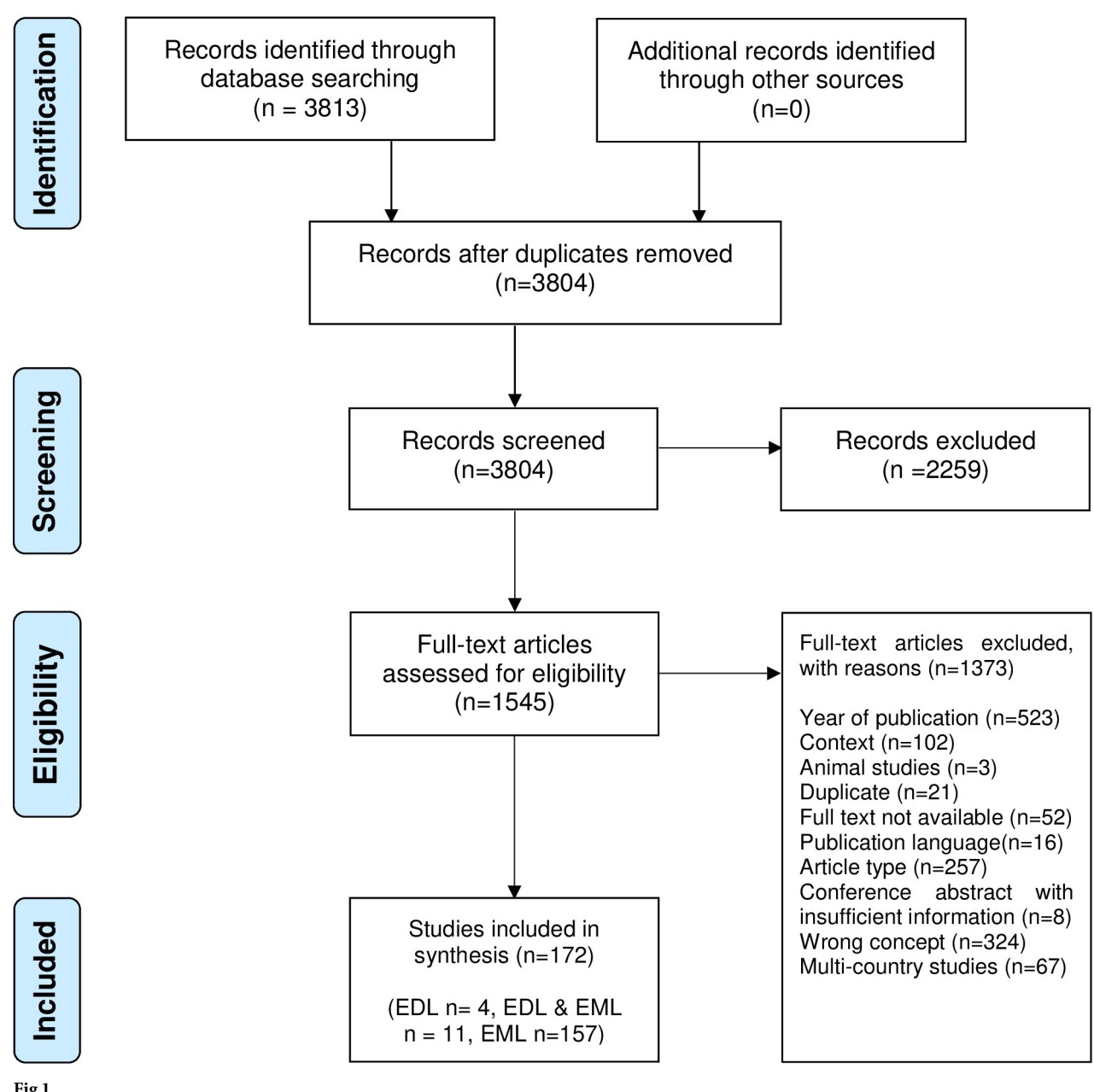

**Fig 1.**

## Characteristics of included studies

We included 172 studies. Four (2.3%) of the studies were on the implementation of EDL, eleven (6.4%) focused both on EDL and EML, and 157 (91.3%) focused only on the EML. A summary of key findings for all included studies can be found in the appendix (S3 Appendix).

Of the (EDL) studies (n = 15), eight (53.3%) were from Eastern Africa, five (33.3%) from Southern Africa, and two (13.3%) from West Africa. Methodologically, twelve (80%) of EDL studies used quantitative methods (cross-sectional designs), and three (20%) used the mixed methods approach (Table 1).

**Table 1. Characteristics of studies about the Essential In Vitro Diagnostics List (EDL).**

| Category | Details | Number (n) |
|---|---|---|
| WHO Essential Diagnostic List (EDL) | Total EDL studies | 15 |
| | EDL sole focus | 4 |
| | EDL and EML combined | 11 |
| Publication year | 2018–2021 | 15 |
| Study Designs | Cross-sectional study | 12 |
| | Mixed methods (Quantitative and Qualitative) | 3 |
| Disease categories | Communicable diseases | 2 |
| | Non-communicable diseases | 12 |
| | Combined communicable & non-communicable diseases | 1 |
| Populations at risk | Adults | 1 |
| | Adults & Children (mixed) | 11 |
| | Unclear | 3 |
| Type of participants* | Health workers | 15 |
| | Patients | 3 |
| | Health care managers | 2 |
| | Laboratory workers | 1 |
| | Other research staff | 1 |
| Region* | Eastern Africa (Kenya, Ethiopia, Madagascar, Tanzania, Uganda, Sudan, Rwanda) | 8 |
| | Southern Africa (South Africa, Namibia, Malawi, Mozambique) | 5 |
| | West Africa (Nigeria, Ghana, Burkina Faso) | 2 |
| Type of facility* | Hospitals | 10 |
| | Primary health care facilities | 7 |
| | Laboratory | 2 |
| | Pharmacies/Dispensaries | 2 |
| | Unclear | 1 |
| Location of health facility | Rural | 1 |
| | Urban | 4 |
| | Rural and urban | 4 |
| | Unclear | 6 |

*Some studies reported more than one detail

Of the EML studies (n = 168), eighty-five (51.2%) were from Eastern Africa, forty-five (27.4%) were from Southern Africa, thirty-two (19.0%) were from West Africa, five (3.0%) from Central Africa and one (0.6%) from North Africa (Table 2).

## Study designs of included studies

Overall, most of the studies (n = 129, 75%) in our review used quantitative study designs, followed by qualitative (n = 26, 15%) and mixed methods (n = 17, 10%). Studies with quantitative methods were mainly descriptive cross-sectional designs (n = 104, 60%), followed by experimental or intervention studies (n = 11, 6%) and cohort study designs (n = 5, 3%). A summary of EDL study designs is presented in Table 1, and EML studies in Table 2.

All studies with a sole focus on EDL were cross-sectional study designs. Most studies on EML were quantitative studies (n = 125, 73%) and primarily descriptive cross-sectional studies.

**Table 2. Characteristics of studies about the Essential Medicine List (EML).**

| Category | Details | Number (n) |
|---|---|---|
| Essential Medicine List (EML) | Total EML studies | 168 |
| | EML focus only | 157 |
| | Combined EML & EDL studies | 11 |
| Publication year | 2010–2021 | 168 |
| Study Designs | Cross-sectional study | 100 |
| | Qualitative study | 26 |
| | Mixed methods | 17 |
| | Cohort study | 5 |
| | Randomized controlled trial | 4 |
| | Interrupted Time Series | 4 |
| | Quasi-Experimental study | 1 |
| | Other Experimental designs | 2 |
| | Other Descriptive designs | 9 |
| Disease categories | Communicable diseases | 19 |
| | Non-communicable diseases | 51 |
| | Both communicable and non-communicable diseases | 51 |
| | Unclear | 47 |
| Populations at risk | Children | 22 |
| | Adults | 16 |
| | Mixed populations (adults and children) | 86 |
| | Unclear | 44 |
| Type of participants* | Patients | 33 |
| | Health workers | 129 |
| | Health care managers | 31 |
| | Laboratory workers | 2 |
| | Community residents | 8 |
| | National Essential Lists Committee | 15 |
| | Health care stakeholders | 13 |
| | Policy document analysis, reviews, Inventory analysis, Ministry officials | 15 |
| | Unclear | 7 |
| Region | North Africa | Egypt (1) |
| | West Africa | Benin, Burkina Faso, Ghana, Mali, Nigeria, Sierra Leone (32) |
| | Central Africa | Cameroon, Gabon (5) |
| | Eastern Africa | Ethiopia, Kenya, Madagascar, Rwanda, Sudan, Tanzania, Uganda (85) |
| | Southern Africa | Botswana, Eswatini, Malawi, Mozambique, Namibia, South Africa, Zambia, Zimbabwe (45) |
| Type of facility* | Hospitals | 95 |
| | Primary Health care facilities | 90 |
| | Pharmacies/Dispensaries | 45 |
| | Community services | 5 |
| | Others (Clinics, Medical centres, National essential list committees) | 4 |
| | Unclear | 13 |

(*Continued*)

**Table 2.** (Continued)

| Category | Details | Number (n) |
|---|---|---|
| Location of health facility* | Rural | 25 |
| | Peri-Urban | 11 |
| | Urban | 26 |
| | Rural and Urban | 36 |
| | Peri-urban and Urban | 6 |
| | Rural, Urban, Peri-Urban | 4 |
| | Unclear | 74 |

*Some studies reported more than one detail

## Quality of evidence

One hundred and sixty-seven (97%) articles were appraised for methodological quality. Fifty-six (33%) articles were graded as having high quality, 52 (30%) as considerable quality, fifty-four (31%) as moderate quality, and 5 (3%) as poor quality. Five (3%) articles were not rated since they did not provide sufficient information to permit a complete MMAT appraisal.

## Synthesis of results

We present key themes about the barriers and enablers of the EDL and EML stratified in SURE themes. Barriers and enablers facing the EDL and EML were similar and were mostly about health system constraints (Fig 2, Tables 3 and 4). The main reported themes across all the SURE domains about barriers facing the EDL were as follows: the health system domain (Poorly equipped/stocked facilities) and socio-political domain (Insufficient legislation/policy and regulations, i.e., lack of policies to facilitate funding allocation for essential tests). The main reported themes across all the SURE domains for barriers facing the EML were as follows: the health system domain (poorly equipped/stocked facilities); the socio-political domain (insufficient legislation/policy and regulation); the provider of care domain (limited knowledge and skills of health workers); the recipient of care domain (poor acceptability of services); Other stakeholders (limited knowledge).

Below we present the most common themes about health systems, social and political level, providers of care, recipients of care, and other stakeholders. The themes about enablers for implementing the EML and EDL are opposite to the health system barriers listed above. They are about tackling the listed barriers and have been summarised in Fig 3.

### Barriers and enablers for the implementation of the EDL and EML

**Health systems-level.** The facility-related constraints were the most reported barriers to implementing WHO essential lists (EDL&EML) (Tables 3 and 4). Unavailability of EDL tests [6, 33–39] and reagent stock-outs [6, 37, 40] were the most prominent themes within the facility-related barriers to EDL implementation. Other EDL barriers referenced lack of proper equipment and supplies described as low availability of key consumables for laboratory diagnosis, limited items of the major laboratory equipment [6, 33], and inadequate infrastructure and space [33] to facilitate laboratory and diagnostics services. Similarly, in the EML implementation, the most prominent themes within this barrier were the low availability and unavailability of essential medicines [41–55].

The enabler themes for the EDL implementation are opposite to the EDL barriers mentioned above [40, 56–58]. The EML enablers are opposite of the EML barriers reported above

| Health Systems Barriers | Essential In Vitro Diagnostics List (EDL) | Essential Medicines List (EML) |
|---|---|---|
| Accessibility of care | Poor financial access<br>Poor geographical access<br>Poor health facility access | Poor financial access<br>Poor geographical access<br>Poor health facility access<br>Poor equitable access |
| Financial resources | Inadequate health facility operational funds<br>High test costs | Insufficient health facility funding<br>Delayed payments from national health agencies |
| Human resource | Shortage of staff<br>Inadequate capacity of laboratory staff | Shortage of staff<br>Inadequate capacity of health workers<br>Low number of specialised healthcare workers |
| Education system | Improper training | Lack of staff training<br>Lack of specialized training<br>Lack of educators to train health workforce |
| Information management systems | Unavailability of operational data | Poor information management practices<br>Poor availability of reports<br>Inadequate information systems<br>Lack of registries |
| Facilities | Unavailability of tests<br>Lack of proper equipment and supplies<br>Inadequate infrastructure | Low availability of essential medicines<br>Unavailability of essential medicines in public and lower-level facilities<br>Inadequate health facility capacity |
| Procurement and distribution systems | Poor forecasting<br>Low inventory levels | Inefficient procurement processes<br>Poor stock management<br>Inefficient distribution systems<br>Poor quantification at facility level<br>Poor training in procurement<br>Inappropriate medicine selection |
| Relationships with norms and standards | Poor availability of guidelines | Unavailability of guidelines<br>Poor availability of EMLs<br>Incompliance to NEMLs and STGs<br>Disconnect between national EMLs and STGs |

**Fig 2.**

and mainly include the availability of essential medicines in facilities [47, 59–89] and adequate capacity of facilities to provide care [71, 90].

 **Social and political level.** The most frequently reported barrier at the social and political level was related to legislation or regulations (Tables 3 and 4). Insufficient policy to facilitate access to essential diagnostics was identified as a barrier to the EDL implementation [34]. Barriers unique to the EML included lack of price regulations or pricing policy [45, 47, 91–95], incompliance to regulations [49, 50, 96–99], lack of structured guidelines for registration and control [59, 67, 100–102]. Other barriers included lack of policies [101, 103, 104], inadequate policies that provide control and use of medicines [105–107], a long registration process [108–

**Table 3. Thematic content analysis of applied SURE codes to barriers to the implementation of EDL.**

| Health system constraints | Proportion of total themes per code |
|---|---|
| Facilities (N = 9, n = 17) | |
| Unavailability of tests | 11 (65%) |
| Reagent stock-outs | 3 (18%) |
| Lack of proper equipment | 2 (12%) |
| Inadequate infrastructure and space | 1 (6%) |
| Accessibility of care (N = 6, n = 8) | |
| Poor financial access | 4 (50%) |
| Poor health facility access | 2 (25%) |
| Poor geographical access | 1 (12.5%) |
| Poorly resourced health facilities | 1 (12.5%) |
| Procurement and distribution (N = 3, n = 3) | |
| Poor supply chain management including poor quantification and low inventory levels | 3 (100%) |
| Human resources (N = 3, n = 3) | |
| Shortage of laboratory staff | 1 (33%) |
| Inadequate number of qualified and skilled lab staff | 1 (33%) |
| General shortage of health worker | 1 (33%) |
| Information systems (N = 3, n = 3) | |
| Absence of clinical case registries | 2 (67%) |
| Unavailability of operational data | 1 (33%) |
| Financial resources (N = 2, n = 2) | |
| Inadequate health facility operational funds | 1 (50%) |
| High test costs | 1 (50%) |
| Education system (N = 2, n = 2) | |
| Improper training of laboratory staff | 1 (50%) |
| Lack of training | 1 (50%) |
| Relationship with norms and standards (N = 1, n = 1) | |
| Poor availability of guidelines | 1 (100%) |
| **Social and Political constraints** | |
| Legislation or regulations (N = 1, n = 1) | |
| Insufficient policy | 1 (100%) |

N = number of studies that cited the SURE theme, n = frequency of applied codes per SURE theme

SURE = Supporting the Use of Research Evidence.

**Table 4. Thematic content analysis of the most applied SURE codes to barriers to the implementation of EML.**

| Health system constraints | Proportion of total themes per code |
|---|---|
| Facilities (N = 76, n = 81) | |
| Low availability of essential medicines in health facilities | 45 (56%) |
| Unavailability of essential medicines in public and lower-level health facilities | 15 (19%) |
| Inadequate health facility capacity | 10 (12%) |
| Human resources (N = 44, n = 62) | |
| Lack of human resource | 19 (31%) |
| Inadequate capacity of health workers | 11 (18%) |
| Low number of specialized healthcare workers | 8 (13%) |
| Accessibility of care (N = 35, n = 53) | |
| Poor financial access | 22 (42%) |
| Poor geographical access | 16 (30%) |
| Poor health facility access | 15 (28%) |
| Procurement and distribution systems (N = 40, n = 49) | |
| Inefficient procurement processes | 16 (33%) |
| Poor stock management | 10 (20%) |
| Inefficient distribution systems | 9 (18%) |
| Relationship with norms and standards (N = 33, n = 35) | |
| Unavailability of guidelines | 16 (46%) |
| Incompliance to guidelines | 8 (23%) |
| Incompliance to national EMLs | 6 (17%) |
| Financial resources (N = 25, n = 25) | |
| Insufficient health facility funding | 22 (88%) |
| Late claimant rebates from national insurance agencies | 2 (8%) |
| Poor effect of financial autonomy | 1 (4%) |
| Information systems (N = 18, n = 18) | |
| Poor information management practices | 7 (39%) |
| Lack of reporting procedures and ordering systems | 4 (22%) |
| Inadequate information systems | 4 (22%) |
| Clinical supervision (N = 11, n = 11) | |
| Lack of supportive supervision | 9 (82%) |
| Inadequate health worker supervision | 2 (18%) |
| Educational system (N = 10, n = 10) | |
| Lack of staff training | 6 (60%) |
| Lack of specialized training | 3 (30%) |
| Lack of educators to train health workforce | 1 (10%) |
| Allocation of authority (N = 10, n = 10) | |
| Limited facility manager authority on facility budget, procurement, pricing, and supply of medicines | 7 (70%) |
| Lack of health workers involvement in drug selection and procurement | 2 (20%) |
| Lack of autonomy by community members and facility managers on the community health fund | 1 (10%) |
| Accountability (N = 8, n = 10) | |
| Lack of accountability of authorities on medicine orders, procurement, distribution, and stock management | 10 (100%) |
| Management and or leadership (N = 9, n = 9) | |
| Inadequate leadership and coordination capacity | 7 (78%) |

(*Continued*)

**Table 4.** (Continued)

| Health system constraints | Proportion of total themes per code |
|---|---|
| Inadequate leadership support to rural-based health workers | 1 (11%) |
| Lack of knowledge and skills by health workers working in managerial roles | 1 (11%) |
| Internal communication (N = 9, n = 9) | |
| Lack of coordination among stakeholders, health managers, and health facilities | 6 (67%) |
| Poor communication of policy change | 2 (22%) |
| Limited communication between health facilities | 1 (11%) |
| Incentives (N = 6, n = 6) | |
| Low motivation of health workers including high workload, poor renumeration, and non-payment of stipends | 4 (67%) |
| Non-payment of suppliers | 1 (16%) |
| Low tax incentives and subsidies to pharmaceuticals | 1 (16%) |
| Bureaucracy (N = 5, n = 5) | |
| Bureaucratic decision making with limited evidence | 5 (100%) |
| Patient flow processes (N = 5, n = 5) | |
| Poor referral practices | 5 (100%) |
| External communication (N = 2, n = 2) | |
| Lack of information/communication material for provision of standard care | 1 (50%) |
| Insufficient levels of essential information for consumers | 1 (50%) |
| **Social and Political constraints** | |
| Legislation or regulations (N = 34, n = 36) | |
| Lack of pricing policy | 10 (28%) |
| Incompliance to regulations | 7 (19%) |
| Lack of structured guidelines | 5 (14%) |
| Donor policies (N = 6, n = 6) | |
| Donors influence on the implementation of EML | 4 (67%) |
| International procurement policies of donors | 1 (17%) |
| Poor policy adoption from the global to national level | 1 (17%) |
| Influential people (N = 5, n = 6) | |
| Pharmaceutical promotions to influence prescription practice and revisions of standard treatment guidelines | 2 (33%) |
| International recommendations | 1 (17%) |
| Lack of support to local pharmaceutical production | 1 (17%) |
| Ideology (N = 4, n = 4) | |
| Market ideologies | 1 (25%) |
| Political ideologies | 1 (25%) |
| Community beliefs and attitudes | 1 (25%) |
| Corruption (N = 4, n = 4) | |
| Corruption practices in public companies | 3 (75%) |
| Trading of counterfeit medicines | 1 (25%) |
| Contracts (N = 3, n = 3) | |
| Absence of national medical contracts | 1 (33%) |
| Delay in contracting pharmaceutical tenders | 1 (33%) |
| Absence of structure contracts | 1 (33%) |
| Political stability (N = 1, n = 1) | |
| Political instability | 1 (100%) |
| **Individual level constraints** | |

*(Continued)*

**Table 4.** (Continued)

| Health system constraints | Proportion of total themes per code |
|---|---|
| **Providers of care** | |
| Knowledge and skills (N = 21, n = 21) | |
| Inadequate training on current evidence-based treatment | 8 (38%) |
| Insufficient number of skilled healthcare workers | 5 (24%) |
| Inadequate providers knowledge on disease management | 3 (14%) |
| Motivation to change or adopt new behaviour (N = 8, n = 10) | |
| Underpayment of healthcare workers | 3 (30%) |
| High workload | 2 (20%) |
| Limited supervision, career, and training opportunities | 2 (20%) |
| Attitudes towards programme (N = 6, n = 6) | |
| Quality concerns of medicines from some manufacturers | 1(17%) |
| Negative attitude towards new quality improvement projects | 1(17%) |
| Poor perceptions and awareness of child-appropriate medicine dosage formulations | 1(17%) |
| **Recipients of care** | |
| Attitudes towards programme (N = 9, n = 10) | |
| Preference for private pharmacies to primary health care centres | 2 (20%) |
| Social cultural influences | 2 (20%) |
| Use of alternative treatments | 2 (20%) |
| Motivation to change or adopt new behaviour (N = 6, n = 7) | |
| Unaffordability of medicines | 2 (29%) |
| Low availability of drugs in health facilities | 1 (14%) |
| Poor access to health facilities | 1 (14%) |
| Knowledge and skills (N = 2, n = 2) | |
| Inadequate consumer knowledge on drugs | 1 (50%) |
| Insufficient understanding of diseases | 1 (50%) |
| **Other stakeholders** | |
| Knowledge and skills (N = 5, n = 5) | |
| Inadequate training on rational use of medicines | 2 (40%) |
| Inadequate knowledge on health commodities and financial report | 1 (20%) |
| Unequal access to information on medical products | 1 (20%) |
| Motivation to change or adopt new behaviour (N = 4, n = 4) | |
| Inadequate drugs and medical supplies | 2 (50%) |
| Low motivation to participate in healthcare programmes | 1 (25%) |
| Poor health worker and patient relationships | 1 (25%) |

N = number of studies that cited the SURE theme, n = frequency of applied codes per SURE theme

SURE = Supporting the Use of Research Evidence

110], and restriction on the use of medicines [111, 112]. Lack of political will in implementing policies [113], lack of a regulatory body for certifying and professionalizing medical and logistical companies [113], and inadequate procedures [114] were also cited as barriers to EML.

The EML enablers reported include supportive health financing policy reforms [111, 115] and the presence of a structured registration process [70] that supported the implementation of the EML.

**Providers of care.** Barriers related to knowledge and skills were the most prominent theme under the providers of care domain. The main barriers identified included inadequate

| Health Systems Enablers | Essential In Vitro Diagnostics List (EDL) | Essential Medicines List (EML) |
|---|---|---|
| Accessibility of care | Financial access<br>Geographical access | Financial access<br>Adequate geographical access<br>Adequate health facility access<br>Access to essential medicines |
| Facilities | Availability of on-site laboratories<br>Availability of tests<br>Adequate facility capacity | Availability of essential medicines<br>Improved infrastructure<br>Adequate facility capacity |
| Procurement and distribution systems | Good inventory levels of tests | Good inventory management<br>Good store management practices<br>Regular distribution systems<br>Lower public procurement prices<br>Utilization of ICT stock management system<br>Efficient supply management |
| Relationships with norms and standards | Availability of diagnosis guidelines | Availability of guidelines<br>Compliance to the essential medicines lists<br>Compliance to treatment guidelines<br>Compliance to policies<br>Compliance to standard operating procedures |

**Fig 3.**

training on current evidence-based treatment [71, 116–119], an insufficient number of skilled healthcare workers [105, 112, 120–122], and inadequate providers' knowledge of disease management [50, 83, 123]. Other barriers included lack of knowledge of inventory management

[124, 125], lack of awareness of available guidelines [52], and poor understanding of partner programmes [106, 126].

The knowledge and skills-related enablers for implementing the EML are opposite to the main barriers listed above [85, 88, 127–129].

**Recipient of care.**   Patients' attitudes regarding acceptability, appropriateness and credibility related to EML implementation were the most prominent theme under the recipient of care. The main barriers reported include a preference to seek care from private pharmacies than primary health care centres due to a lack of essential drugs and beliefs on the quality of medicines [101, 108], social-cultural influences [130, 131], and use of alternative treatments (traditional medicine) [101, 108]. Other barriers reported include perception of the unaffordability of drugs [132], uncertainty on availability of services [133], drug safety concerns [134], and low health-seeking behaviour [132].

**Other stakeholders** *(community health committees, community leaders, programme managers, donors, policymakers, opinion leaders).*   The most prominent theme in this domain was the stakeholder's knowledge and skills to facilitate the implementation of EML. The common barriers reported include inadequate training on rational use of medicines [135, 136], inadequate understanding of health commodities and financial reports amongst health facility and governing committee members [89], unequal access to information on medical products to all stakeholders [98], and variation in the knowledge of child-appropriate dosage formulations among stakeholders [50].

Stakeholders' knowledge and skills on the quality, safety, and efficacy of medicines and pharmacoeconomic evaluations in selecting medicines for the EML were cited as enablers for implementing the EML [137].

## Discussion

This scoping review was conducted to map evidence on implementing the WHO's essential lists in Africa to guide the effective implementation of the new WHO EDL. Our comprehensive scoping review identified themes based on the SURE framework into the barriers and enablers for implementing WHO essential lists across 172 articles. In lieu of the novelty of the EDL, there was limited published primary research on the implementation of WHO EDL. We found many studies reporting evidence on the implementation of EML in Africa. The review findings showed that the main barrier facing the more established EML, and newly introduced EDL was poorly equipped health facilities that entailed unavailability of essential in vitro diagnostics and medicines, stock-outs of laboratory reagents, and inadequate infrastructure and space to enable health service delivery. The main reported enabler was facility-related; considerable availability of basic tests and medicines and improved facility capacity to provide essential services.

Most of the studies in our review used quantitative methods, with nearly two-thirds of all studies using cross-sectional study designs. There were few qualitative studies, mixed-methods studies, and randomized trials. Qualitative studies are useful for exploring and understanding barriers and facilitators for the EDL and EML in different contexts [138]. Experimental studies are more useful for evaluating interventions that may improve the effectiveness of the EDL on health outcomes [139].

Similar work to ours, a systematic review by Peacocke et al., [140] explored the process of adapting the WHO EML at the national level. The authors provided key insights on the complexities and interdependencies essential to implementing the EML. Their review focused on key factors influencing the adaptation and implementation process of the EML at the macro level of the health system: country-level institutional structure; legislative and regulatory

frameworks; governance, leadership, and coordination for NEMLs. Our review provides further insights and maps evidence on implementing the WHO EDL and EML at national levels, focusing on the African context. In our review, we present barriers and enablers facing the EDL and EML at different levels of implementation; individual, health system, and social and political levels that influence the implementation of the WHO essential lists in-depth.

The essential lists and, more recently, the EDL alone are insufficient to ensure their impact on access and health outcomes. A sound health system is vital to strengthen the existence of the lists. In Africa, health systems face complex challenges such as the continued burden of communicable and non-communicable diseases pandemics amidst limited resources [141–143]. Indeed, our review highlighted that health systems constraints remain the main barrier to implementing the EDL and EML. Such barriers included poorly equipped health facilities with limited essential tests and medicines available. Socio-political constraints, such as the inadequacy of existing legislation and regulations to support the implementation of EDL and EML, also impact other domains. They influence the health system's performance and, subsequently, limit the capacity of providers of care and other stakeholders and acceptability and service availability to care recipients, as highlighted by our findings. Many influencing factors in the health system determine the access, implementation, and effectiveness of diagnostic tests. Dealing with such challenges requires that decisions on health systems are informed by robust evidence that applies to the local context. Policymakers and health decision-makers can look to evidence-informed approaches, especially synthesising health policy and systems evidence and contextualize findings to their settings. Methods for conducting or utilizing health systems synthesis can be found in the WHO methods guide for evidence synthesis for health policy and systems [144].

Evidence-informed approaches are useful in guiding the adapting process and improving the implementation of the lists. WHO has a guidance resource enabling African countries to adopt the WHO EDL to national contexts [145]. To our knowledge, in Africa, Nigeria is the only country that has adopted the WHO EDL list and developed its own national EDL [146]. Many African countries have adopted the WHO EML in national settings. Thirty-nine of the 47 countries in the WHO Africa region have developed NEML linked to STGs [147]. However, stock-outs and limited access to medicines persist, emphasising the importance of enabling health systems to strengthen the implementation of the essential lists and ensure their impact. Evidence about the evidence-informed approaches or processes in adapting the EML has been published by South Africa [137], Ghana [148], and Tanzania [70]. These publications highlighted enablers such as a well-structured and rigorous process [137, 148], utilization of evidence summaries in decision-making [137, 148], and involvement of a diverse committee and stakeholder engagements [137, 148]. Challenges included insufficient and intermittent funding [148], limited use of scientific evidence [70], lack of expertise in evidence synthesis [70, 148] and health economic analyses [70, 137] in the review and development of NEML. Besides providing adaption guides for the essential lists, the implementation handbook guides can be released in conjunction with the versions of the model lists.

The WHO also released a handbook for monitoring the building blocks of health systems. It is structured around the six main building blocks of the WHO health systems framework: service delivery, health workforce, health information systems, access to essential medicines, financing, and leadership and governance [149]. The proposed measures of health systems performance are crucial in health systems strengthening and valuable tools to accurately monitor the health system's progress across the six building blocks over time. It facilitates the development of a sound Country monitoring strategy providing an enabling environment and sustainable scale-up of governance tools such as the EML and the newly introduced EDL. The EML and EDL play a vital role in realising UHC and access to quality health service delivery

[150]. The impact of the essential medicine and diagnostics lists will become truly effective only in well-functioning strengthened health systems. The core indicators to performance measures of key building blocks, including access to essential medicines and technologies, health service delivery, health workforce, health information systems, health financing, and leadership and governance [149], are all critical to the development, review, and implementation of the essential lists. The use of core indicators in the health systems could also assist in addressing EDL and EML implementation barriers timely, efficiently, and effectively to impact populations' health outcomes.

In this review, there were notable successes of interventions developed to address barriers to the EML implementation that could be considered useful in the EDL implementation. The RDF [61, 89, 151], PBF [119, 152, 153], and P4P [154, 155] interventions addressed several barriers to implementing the EML: accessibility for care-related barriers, facility-related barriers, incentive-related barriers, information system-related barriers, accountability-related barriers, and facility financial resource-related barriers. The revolving fund pharmacy (RFP) [156], accredited drug dispensing outlets (ADDOS) [128], and auditable pharmaceutical services and transaction system (APTS) [157] interventions also addressed the facility-related barriers. They contributed to the improved availability of essential medicines. Procurement and distribution-related barriers were addressed through direct distribution of supplies from partners [73, 125, 158–160], PBF [161], RDF programmes [61, 89, 151], and utilization of ICT [108, 162] in stock management. Similar interventions could be used to address the shortfalls of the EML and strengthen the EDL implementation designs. However, considering the country's context and specificities to be addressed will be crucial when implementing interventions. Some interventions worked in some contexts and did not work in other contexts. For instance, the PBF intervention did not affect the stock-out rate of essential medicines compared to payments not tied to the performance of essential medicines in some contexts [164]. On the other hand, the provision of financial incentives in the P4P intervention addressed some health system barriers; still, it was reported to have no evidence for increasing healthcare workers' motivation [154]. Though financial and non-financial incentives may motivate implementation, they can unrealistically raise expectations and hinder implementation in the long run due to sustainability issues [163].

We evaluated the existing literature through a systematic and rigorous process that involved reviewing qualitative, quantitative, and mixed methods studies using established guidance for scoping reviews. To inform the implementation of EDL, we also referred to a representative sample of the established EML. We did not include non-English studies; hence, we could have missed studies published by French, Portuguese, or Arabic-speaking African countries. Secondly, due to accessibility limitations, we excluded 52 EML articles and multi-country studies about EML (n = 67) due to the vast number of full-text EML studies and the high likelihood of data saturation given rich, in-depth information from single countries in the multiple numbers of available studies. We also did not explore the process of adapting the WHO essential list to national contexts. Trend analysis from EML inception to the date of implementation aspects of the EML would help identify the successes, pitfalls, and plateaus of EML implementation over four decades. However, this was out of the scope of this work.

There has been limited primary research published on essential in vitro diagnostics in Africa since the introduction of the WHO EDL in 2018. Further studies can be conducted to provide contextual insights on the capacity of health systems to support the successful implementation of national EDLs bearing in mind the need to improve access to essential in vitro diagnostics in Africa. Consideration of dissemination and implementation frameworks such as the CFIR (Consolidated Framework for Implementation Research) [164, 165], RE-AIM (Reach, effectiveness, adoption, implementation, and maintenance) [165, 166], and PRISM

(practical, robust implementation sustainability model) [167, 168] frameworks would be crucial when planning the implementation of the essential lists to guide adoption, adaptation, and evaluation of the lists. Qualitative research and process evaluations can be done to evaluate the impact of the essential lists and identify enablers and challenges to their implementation. More implementation trials or experimental studies can be conducted to assess effective interventions in different settings.

## Conclusion

The most dominant constraints facing EML implementation, a more established WHO essential list and the new EDL are mainly about the health system. The main theme barrier was poorly equipped health facilities, including limited availability of essential in vitro diagnostics and medicines and stock-outs, which mainly limited the implementation of the EML and EDL. The EDL implementation can learn from interventions to improve the availability and supply of essential medicines. When developing and implementing the National EDLs, consideration of these barriers will strengthen health service delivery, access to essential diagnostics and universal health coverage. Financial and non-financial incentives may be enablers, but their effect varies in different contexts. Most of the EDL and EML studies used cross-sectional designs. While cross-sectional designs are suitable for identifying implementation challenges, exploratory qualitative and interventional designs are more suitable for understanding the challenges and evaluating the impact and effectiveness of the EDL and EML interventions and health outcomes, respectively.

## Supporting information

**S1 Checklist. PRISMA-ScR checklist.**
(DOCX)

**S1 Table. Barriers to the implementation of an essential diagnostic and medicines list.**
(DOCX)

**S2 Table. Enablers for the implementation of an essential diagnostic and medicines list.**
(DOCX)

**S1 Appendix. Deviations from protocol.**
(DOCX)

**S2 Appendix. Search strategy and SURE framework checklist.**
(DOCX)

**S3 Appendix. Summary of included studies.**
(DOCX)

## Acknowledgments

We thank Vittoria Lutje, Information Specialist at the Liverpool School of Tropical Medicine, who advised on and developed the search strategy. The authors express their gratitude to the Kenya Medical Research Institute for the support provided for this review.

## Author Contributions

**Conceptualization:** Moriasi Nyanchoka, Mercy Mulaku, Eleanor Ochodo.

**Data curation:** Moriasi Nyanchoka, Mercy Mulaku, Eleanor Ochodo.

**Formal analysis:** Moriasi Nyanchoka, Eleanor Ochodo.

**Funding acquisition:** Eleanor Ochodo.

**Investigation:** Moriasi Nyanchoka, Mercy Mulaku, Bruce Nyagol, Eddy Johnson Owino, Eleanor Ochodo.

**Methodology:** Moriasi Nyanchoka, Mercy Mulaku, Eleanor Ochodo.

**Project administration:** Eleanor Ochodo.

**Resources:** Eleanor Ochodo.

**Supervision:** Eleanor Ochodo.

**Validation:** Moriasi Nyanchoka, Eleanor Ochodo.

**Visualization:** Moriasi Nyanchoka, Eleanor Ochodo.

**Writing – original draft:** Moriasi Nyanchoka, Eleanor Ochodo.

**Writing – review & editing:** Moriasi Nyanchoka, Mercy Mulaku, Bruce Nyagol, Eddy Johnson Owino, Simon Kariuki, Eleanor Ochodo.

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
