## [Decision Letter · Decision Letter 0]

9 Sep 2022

PGPH-D-22-01071

Implementing essential diagnostics-learning from essential medicines: A scoping review

Dear Dr. Nyanchoka,

Thank you for submitting your manuscript to PLOS Global Public Health. After careful consideration, we feel that it has merit but does not fully meet PLOS Global Public Health’s publication criteria as it currently stands. Therefore, we invite you to submit a revised version of the manuscript that addresses the points raised during the review process.

We look forward to receiving your revised manuscript.

Kind regards,

Ejemai Eboreime, MD, MSc, PhD

Academic Editor

Journal Requirements:

1.We have amended your Competing Interest statement to comply with journal style. We kindly ask that you double check the statement and let us know if anything is incorrect. 

2.The resolution of Figure 2 is very low and somewhat difficult to read. It is important that our Editors and Peer Reviewers are able to read all parts of a submission. Please replace these figures with higher resolution copies.

Additional Editor Comments (if provided):

This article is said to be a scoping review. However I have very serious concerns about the article, in addition to the reviewers' concerns.

First the objective of this article is unclear. The authors state on lines 397-399 that : "The objective of this scoping review was to map evidence on the implementation and evaluation of the WHO’s essential lists in African countries to guide the effective implementation of the new WHO EDL."

This objective is not SMART and would therefore make it difficult for peer reviewer to provide focused review of the article. It is unclear what the authors mean by "evidence on the implementation and evaluation". Further the authors conducted a thematic analysis of the data but failed to describe in detail how this was done.

It is strange that the authors conducted a quality check in a scoping review. This is normally within the purview of systematic reviews, not scoping reviews. Even though the authors claim to have been guided by the PRISMA-ScR checklist, there is no convincing evidence that this is the case, judging from the way the report is written.

The results way the results are presented are not only unconventional, they are confusing. The authors present several results tables in the form of frequency tables. One is unable to tell which teams are relevant or not, given the the objectives and scope of this review are unclear. Typically results of scoping reviews are presented by listing each article with their key findings guided by the pre-determined themes. Perhaps the authors may want to read a few previously published scoping review articles to guide them. This link may also be helpful: https://guides.library.ontariotechu.ca/c.php?g=723749&p=5180372#:~:text=The%20results%20of%20a%20scoping,and%20research%20methods%3B%20and%2For

The article is also unnecessarily long. The article can be presented in a significantly shorter better written manuscript.

Reviewers' comments:

Reviewer's Responses to Questions

**Comments to the Author**

1. Does this manuscript meet PLOS Global Public Health’s publication criteria? Is the manuscript technically sound, and do the data support the conclusions? The manuscript must describe methodologically and ethically rigorous research with conclusions that are appropriately drawn based on the data presented.

Reviewer #1: Partly

Reviewer #2: Partly

2. Has the statistical analysis been performed appropriately and rigorously?

Reviewer #1: No

Reviewer #2: N/A

3. Have the authors made all data underlying the findings in their manuscript fully available (please refer to the Data Availability Statement at the start of the manuscript PDF file)?

Reviewer #1: Yes

Reviewer #2: Yes

4. Is the manuscript presented in an intelligible fashion and written in standard English?

Reviewer #1: No

Reviewer #2: Yes

5. Review Comments to the Author

Reviewer #1: Authors have described in about 100 pages every finding they found in health services. They have used EML and EDL and then related everything to these two lists. It is not clear as to how WHO EML and EDL are meant to be known by providers unless there is national EML and EDL. This is the first finding to be given. How many countries have national EML and EDL based on WHO guidance. Then there should be a check on standard treatment protocols where these national EML and EDL elements are listed. Only then the service provision can be compared.

Reviewer #2: The manuscript highlights various challenges in implementing essential diagnostic and essential medicine list in Africa. The methodology for data extraction is clearly defined. There are a few suggestions

Major

1. The manuscript has a bit of repetitive pattern in the result section. The barriers listed under different headings such as individual level, health system level and the policy level broadly include similar issues. In my opinion it will be better to club them together instead of describing separately. For example similar challenges are described in provider level challenges and health system in terms of human resource, lack of knowledge, lack of training and guideline knowledge.

If feasible identify broad headings and describe to avoid repetition.

2. The discussion should be restructured to focus on the key findings from the study . It would be helpful to evaluate the differences between countries which have implemented EDL (Nigeria)

Minor

1.The methodology had mentioned that selection of studies for EML was restricted to those conducted in single country. However, the table has described it region wise.

2.Some of the items in table 3 and 4 do not add to 100%

3.Please provide full form of ICT in the beginning

6. PLOS authors have the option to publish the peer review history of their article (what does this mean?). If published, this will include your full peer review and any attached files.

**Do you want your identity to be public for this peer review?** For information about this choice, including consent withdrawal, please see our Privacy Policy.

Reviewer #1: No

Reviewer #2: **Yes: **Manju Sengar

---

## [Decision Letter · Decision Letter 1]

18 Nov 2022

Implementing essential diagnostics-learning from essential medicines: A scoping review

PGPH-D-22-01071R1

Dear Mr Nyanchoka,

We are pleased to inform you that your manuscript 'Implementing essential diagnostics-learning from essential medicines: A scoping review' has been provisionally accepted for publication in PLOS Global Public Health.

Best regards,

Ejemai Eboreime, MD, MSc, PhD

Academic Editor

Reviewer Comments (if any, and for reference):

Reviewer's Responses to Questions

**Comments to the Author**

1. If the authors have adequately addressed your comments raised in a previous round of review and you feel that this manuscript is now acceptable for publication, you may indicate that here to bypass the “Comments to the Author” section, enter your conflict of interest statement in the “Confidential to Editor” section, and submit your "Accept" recommendation.

Reviewer #1: All comments have been addressed

Reviewer #2: All comments have been addressed

2. Does this manuscript meet PLOS Global Public Health’s publication criteria? Is the manuscript technically sound, and do the data support the conclusions? The manuscript must describe methodologically and ethically rigorous research with conclusions that are appropriately drawn based on the data presented.

Reviewer #1: Yes

Reviewer #2: Yes

3. Has the statistical analysis been performed appropriately and rigorously?

Reviewer #1: Yes

Reviewer #2: N/A

4. Have the authors made all data underlying the findings in their manuscript fully available (please refer to the Data Availability Statement at the start of the manuscript PDF file)?

Reviewer #1: Yes

Reviewer #2: Yes

5. Is the manuscript presented in an intelligible fashion and written in standard English?

Reviewer #1: Yes

Reviewer #2: Yes

6. Review Comments to the Author

Reviewer #1: Thank you for the revision. Capacity of the health system is critical and this is context driven. One way to make this more relevant is to analyze the availability by a socio-economic grouping. HDI index of countries can be a good one. Use this stratification and see the availability to use EML and EDL in different groups. https://en.wikipedia.org/wiki/List_of_African_countries_by_Human_Development_Index

Reviewer #2: None

7. PLOS authors have the option to publish the peer review history of their article (what does this mean?). If published, this will include your full peer review and any attached files.

**Do you want your identity to be public for this peer review?** For information about this choice, including consent withdrawal, please see our Privacy Policy.

Reviewer #1: No

Reviewer #2: **Yes: **Manju Sengar
